# Peer review of "Improving Factuality by Contrastive Decoding with Factual and Hallucination Prompts"

_sensors, 2024, doi:10.3390/s24217097_

Round 1
Reviewer 1 Report
Comments and Suggestions for Authors
Summary
This paper proposes a novel decoding strategy called DFHP (Decoding by Factual and Hallucination Prompts) to mitigate hallucinations in large language models. It combines prompt engineering with contrastive decoding to enhance factuality in model outputs without additional training. The proposed method is evaluated using multiple-choice and open-ended generation tasks, showing improvements in factual accuracy across various datasets.
Strengths
- The paper addresses an important issue in natural language processing, namely hallucinations in large language models, with a well-motivated solution.
- The proposed DFHP method does not require additional training, which makes it computationally efficient.
- The experimental results demonstrate clear improvements in factual accuracy, validated on a variety of tasks, including multiple-choice and chain-of-thought reasoning datasets.
Weaknesses
- The novelty of the work is not clearly articulated in the introduction, making it difficult for readers to grasp the specific contribution compared to existing methods.
- The limitations of the proposed method, especially regarding its reduced informativeness in open-ended tasks, are not adequately discussed.
Detailed Comments
1. The introduction lacks a clear statement of novelty. While the proposed method of contrastive decoding using factual and hallucination prompts is an interesting idea, it is not explicitly compared to other existing techniques in the introduction. The paper should better highlight how DFHP advances beyond similar approaches like context-aware decoding or other hallucination-mitigation strategies.
2. The experimental setup, including dataset specifications and parameter choices, is missing key details. The reader is left wondering about the exact configurations of the multiple-choice and open-ended tasks. It would be beneficial to include more comprehensive information on the dataset splits, evaluation metrics, and any hyperparameter tuning performed.
3. The paper is well-structured with a comprehensive review of relevant literature, but its scope and impact could be broadened by citing these works: Privacy-preserving Fine-grained Data Sharing with Dynamic Service for the Cloud-edge IoT; PkT-SIN: A Secure Communication Protocol for Space Information Networks with Periodic k-Time Anonymous Authentication; An Efficient Privacy-aware Split Learning Framework for Satellite Communications; Privacy-preserving Universal Adversarial Defense for Black-box Models; It's All in the Touch: Authenticating Users with HOST Gestures on Multi-Touch Screen Devices}; PonziGuard: Detecting Ponzi Schemes on Ethereum with Contract Runtime Behavior Graph (CRBG); Vulseye: Detect Smart Contract Vulnerabilities via Stateful Directed Graybox Fuzzing
4. The paper omits a critical discussion of the limitations of the DFHP method. For instance, the trade-off between factual accuracy and informativeness is touched upon but not thoroughly explored. The authors should discuss scenarios where their method might underperform, such as tasks requiring a high degree of creativity or open-ended responses.
5. In the open-ended generation tasks, the paper reports a reduction in informativeness due to frequent use of "I have no comment" as a safe fallback response. This is a notable drawback, especially in creative or exploratory text generation. While the factual accuracy improves, the reduction in content quality may render the method less useful in certain applications. This needs more emphasis in the conclusions and discussion sections.
6. The results on larger models (e.g., LLaMA-13B) are mentioned but not thoroughly analyzed. It would strengthen the paper to discuss whether the proposed method scales effectively to even larger models, like GPT-4, and how the scaling affects both factuality and informativeness metrics. A deeper exploration of how model size interacts with the DFHP method would be beneficial.
7. The paper briefly explores different prompt designs but stops short of a detailed investigation into more sophisticated prompting techniques like chain-of-thought or role-based prompting. A more thorough exploration of the impact of prompt design choices on model performance would add depth to the method section.
Comments on the Quality of English Languagesee report
Reviewer 2 Report
Comments and Suggestions for Authors
you must to Clarity of Problem Definition and further elaborating on how it differs from existing strategies in a practical sense would strengthen the motivation for the research.
.must Explain retrieval-augmented generation (RAG) in simpler terms how the contrastive nature helps filter out hallucinations more effectively than these other techniques.
need to add a detailed discussion of the potential biases or limitations of human evaluation should be included. How consistent were the human evaluators? Could there be any subjectivity in their judgment, particularly when assessing borderline cases of hallucinations?
Discuss how future work might balance factuality and informativeness. Could there be a threshold for acceptable refusals, or is there a way to fine-tune the system to encourage more confident but accurate responses?
must better showcase the model’s real-world applicability.
Adding one or two additional case study examples in highly specialized domains would enhance the impact of this section.
Round 2
Reviewer 2 Report
Comments and Suggestions for Authors
To improve the clarity and depth of the paper, consider the following revisions:
Abstract: Specify the specific applications of the proposed method and provide a more comprehensive explanation of the experimental results to offer broader context and relevance.
Introduction: Expand the introduction to clearly define the problem statement and highlight the significance of addressing it.
Related Works: Provide a more thorough discussion of related works, ensuring it covers relevant research in greater depth.
Methodology: Expand and clarify the methodology section, including a detailed explanation of the methods, treatments, and procedures used.
Tables: Improve the presentation of tables by ensuring results are clearly displayed and providing properly formatted, descriptive titles for clarity.
Results Section: Clarify the results section with a detailed explanation to enhance understanding and interpret the findings effectively.
Formatting: Adjust the numbering and titles throughout the paper to maintain consistency and coherence.
Conclusion: Refine and further develop the conclusion to enhance its clarity, impact, and summarization of key findings.
